# Analysis of Oral Microbiota in Elderly Thai Patients with Alzheimer’s Disease and Mild Cognitive Impairment

**DOI:** 10.3390/ijerph21091242

**Published:** 2024-09-20

**Authors:** Narongrit Sritana, Atitaya Phungpinij

**Affiliations:** Molecular and Genomics Research Laboratory, Centre of Learning and Research in Celebration of HRH Princess Chulabhorn’s 60 th Birthday Anniversary, Chulabhorn Royal Academy, Bangkok 10210, Thailand; atitayaphungpinij@gmail.com

**Keywords:** Alzheimer’s disease, dementia, neurobiology, oral, microbiome, long read next-generation sequencing

## Abstract

Alzheimer’s disease (AD) is a neurodegenerative disease that predominantly affects the older adult population. Neuroinflammation may be triggered by the migration of oral microbiota composition changes from the oral cavity to the brain. However, the relationship between oral microbiota composition and neurodegenerative diseases, such as AD, remains poorly understood. Therefore, we conducted a comprehensive comparison of the relative abundance and diversity of bacterial taxa present in saliva among older adults diagnosed with AD, those with mild cognitive impairment (MCI), and healthy controls. Saliva samples and clinical data were collected from 10 AD patients, 46 MCI patients, and 44 healthy older adults. AD patients had lower Clinical Dementia Rating, Montreal Cognitive Assessment, and Mini-mental Status Examination scores, and induced microbial diversity, than the MCI and control groups. Moreover, AD patients exhibited significantly higher levels of Fusobacteriota and Peptostreptococcaceae and lower levels of Veillonella than the MCI and control groups. In conclusion, a high abundance of Fusobacteria at various levels (i.e., phylum, class, family, and genus levels) may serve as a biomarker for AD. The analysis of oral microbiota dysbiosis biomarkers in older adults may be valuable for identifying individuals at risk for AD.

## 1. Introduction

Alzheimer’s disease (AD) is a complex neurodegenerative disorder that affects the limbic and association cortices of the brain. It is characterized by the progressive degeneration of neurons, which leads to cognitive decline and memory impairment. AD is a major public health issue, with approximately 44 million people worldwide affected by the disease [1,2]. The hallmark symptoms of AD are memory impairment and cognitive decline, which are the primary drivers of clinical diagnosis [3]. The pathophysiology of AD involves the accumulation of two major proteinopathies in the brain: amyloid beta (Aβ) and tau [4]. Aβ is derived from the amyloid beta-protein precursor and forms plaques in the brain [1]. In contrast, tau is a microtubule-associated protein that forms neurofibrillary tangles [5]. Both Aβ and tau are thought to contribute to the progression of AD and have thus been considered potential biomarkers of AD [4]. Genetic factors also play a crucial role in the development of AD. Several genes that are definitively linked to inherited forms of the disease have been identified. These genes have been shown to increase the production and deposition of Aβ in the brain, which leads to altered intramembranous cleavage of the Aβ precursor protein [6]. Understanding the genotype-to-phenotype relationships in familial AD has provided valuable insights into the mechanisms underlying the disease and opened up new avenues for the development of therapeutic interventions. Furthermore, recent research has highlighted the importance of the genetic background to the progression of tau pathology in AD. Different mouse strains exhibit variations in tau spread and accumulation, which suggests that genetic factors contribute to the rate of disease progression [5,7]. This finding has implications for understanding patient-to-patient variability in the rate of disease progression and may help provide targets for future research and treatment strategies.

Mild cognitive impairment (MCI) is a transitional state between normal cognitive aging and AD. It is characterized by cognitive decline greater than what would be expected for an individual’s age but that does not significantly impair daily functioning [8]. Because individuals with MCI have a higher risk of progressing to AD than those without MCI, MCI is considered a prodromal stage of AD [9]. The diagnosis of MCI is based on clinical criteria, which include cognitive and functional ability assessments [10]. Various criteria for MCI diagnosis have been proposed by research groups: the International Working Group-1, International Working Group-2, and National Institute of Aging-Alzheimer Association criteria. These criteria aim to identify individuals who are at a higher risk of developing AD based on cognitive impairment and biomarkers [9]. Biomarkers, such as amyloid and neuronal injury markers, can help identify individuals with MCI who are more likely to progress to AD.

Although the exact cause of MCI and AD remains unclear, emerging evidence suggests that chronic inflammation and immune dysregulation play a role in the pathogenesis of the disease. In recent years, there has been growing interest in the potential link between oral health and AD, particularly the role of oral microbiota in the development and progression of the disease [11]. The oral microbiome is a complex ecosystem of microorganisms that inhabit the oral cavity. It plays a crucial role in maintaining oral health and has been implicated in various systemic diseases, such as cardiovascular disease, diabetes, and respiratory infections [12]. Recent studies have reported a potential association between oral microbiota and AD. For example, one study found that elevated levels of antibodies against periodontal bacteria and tumor necrosis factor-alpha (TNF-α) are associated with AD [11]. TNF-α is a pro-inflammatory cytokine that plays a key role in the immune response [13]. The presence of elevated antibodies against periodontal bacteria and increased TNF-α levels may indicate chronic inflammation and immune activation, which have been implicated in the pathogenesis of AD. Another study investigated the effects of supplementing infant formula with *Lactobacillus rhamnosus* GG (a probiotic bacteria) on atopic eczema and cow’s milk allergy. They did not directly examine the relationship between oral microbiota and AD; however, the study highlights the therapeutic potential of probiotic bacteria in the management of immune-related diseases [14]. In recent years, dietary interventions targeting the gut microbiome as a strategy for preventing or managing cognitive decline have garnered increased attention. For example, one study found that the modified Mediterranean–ketogenic diet modulated the gut microbiome and short-chain fatty acid production in individuals with MCI [15]; moreover, AD markers showed improvements. This suggests that dietary interventions have the potential to modify the gut microbiome and improve cognitive function in individuals with MCI. Given the link between immune dysregulation and MCI or AD, further research exploring the preventative and therapeutic effects of probiotics on MCI and AD is warranted.

In addition, the compositions of both the gut and oral microbiomes can be significantly influenced by dietary changes. Research has shown that alterations in diet can lead to shifts in the diversity, composition, and stability of the gut microbiota [16]. For instance, major dietary shifts, such as the increased intake of processed carbohydrates, have been associated with significant changes in the diversity and composition of the oral microbiome [17]. Moreover, dietary interventions have been found to impact the gut microbiome, with changes in the gut microbiome composition explaining a portion of the variance in serum metabolites in response to diet [18].

The study of the human microbiome—the collective genomes of microorganisms residing in and on the human body—has revolutionized our understanding of the role of microorganisms in health and disease [19]. Traditional sequencing methods have offered valuable insight into the composition and diversity of microbial communities. However, these methods are unable to accurately capture the full genetic potential of the microbiome. To address this limitation, third-generation sequencing technologies have emerged as powerful tools for studying the microbiome, enabling longer read lengths and improved resolution of complex microbial communities [20]. Third-generation sequencing technology, such as PacBio and Oxford Nanopore sequencing, offers advantages over traditional sequencing methods by generating longer reads, which enable the assembly of complete genomes and the identification of novel microbial species [21]. Furthermore, these methods provide a more comprehensive view of the functional potential of the microbiome by capturing full-length genes and identifying gene variants. The application of third-generation sequencing to the study of the microbiome has yielded significant insight into the genetic diversity and functional potential of microbial communities. For example, metagenomic sequencing was used to establish a comprehensive gene catalog of the human gut microbiome. This catalog, derived from fecal samples of European individuals, contained approximately 3.3 million non-redundant microbial genes, providing a wealth of information about the genetic potential of the gut microbiome. Moreover, the study highlighted the vast genetic diversity within the microbiome and identified genes associated with various health conditions, such as obesity and inflammatory bowel diseases [20].

Collectively, emerging evidence suggests associations among oral microbiota, inflammation, and AD. Furthermore, chronic inflammation and immune dysregulation may contribute to the pathogenesis of MCI and AD, and oral microbiota may be involved in the modulation of these processes [22]. However, further research is needed to elucidate the underlying mechanisms and establish the clinical implications of oral microbiota in MCI and AD. Understanding the link between oral health and AD may open new avenues for early detection, prevention, and treatment strategies for AD. In this study, we compared the microbiota diversity among AD patients, MCI patients, and healthy people with the aim of providing data to support the management of patients with AD, especially Thai patients.

## 2. Materials and Methods

### 2.1. Sample Collection

This study was approved by the Ethics Committee of Human Research of Chulabhorn Research Institute (EC No. 043/2560). Saliva samples were collected from 100 older adult native Thai participants who had visited Chulabhorn Hospital, Ramathibodi Hospital and King Chulalongkorn Memorial Hospital in the project of Holistic approach of Alzheimer’s disease in Thai people (HADThai study). The OMNIgene^®^ ORAL collection kit (DNA Genotek, Ottawa, ON, Canada) was used to collect saliva samples according to manufacturer instructions. Samples were stored at room temperature until DNA extraction was performed. Patients were diagnosed with AD or MCI by neurologists according to the National Institute of Neurological and Communicative Disorders and Stroke and the Alzheimer’s Disease and Related Disorders Association criteria [23]. 

### 2.2. DNA Extraction

To prepare the sample for DNA extraction, a microbial standard (ZymoBIOMICS™ Spike-in control II, Zymo Research, Irvine, CA, USA) was completely thawed on ice and mixed thoroughly using a vortex. We then added 40 μL of spike-in control to 500 μL of the saliva sample and incubated at 4 °C overnight. DNA was then extracted using ZymoBIOMICS™ DNA Miniprep Kit (Zymo Research, Irvine, CA, USA) extraction with adaptation. First, 540 μL of the prepared sample was added to 500 μL of the lysis solution and beads (size = 0.1 and 0.5 mm) using a 1.5mL microtube. The tubes were fastened to a Disruptor Genie vortex (Scientific Industries, Bohemia, NY, USA) using a vortex adapter. The samples were vortexed at maximum speed for 20 min. The samples were then centrifuged at 10,000× *g* for 1 min, and 400 μL of supernatant was transferred to an III-F filter and centrifuged again at 8000× *g* for 1 min. Subsequently, 1200 μL of DNA binding buffer was added to the filtrate, and the mixture was transferred to an IICR column and centrifuged at 10,000× *g* for 1 min. Once the flow-through was discarded, the column was washed with 400 μL of Wash Buffer 1, 700 μL of Wash Buffer 2, and 200 μL of Wash Buffer 2. The column was then centrifuged at 10,000× *g*, and the flow-through was discarded. An additional 200 μL of Wash Buffer 2 was added to the column and centrifuged for 10,000× *g* for 1 min. For the elution process, 50 μL of deoxyribonuclease/ribonuclease-free water was added to the column, incubated for 3 min, and centrifuged at 10,000× *g* for 3 min. Finally, the DNA eluate was passed through the III-HRC prepped filter and centrifuged at 10,000× *g* for 3 min. The elution step was repeated with incubation for 3 min at room temperature before centrifugations.

### 2.3. Library Preparation and Sequencing

To examine the diversity of bacterial communities present in the saliva samples, we used the SMRT PacBio sequencing technology, which is a sequencing method that allows the generation of long-read sequencing. We first amplified the full-length 16 S ribosomal DNA (rDNA) sequence using a pair of bacteria-specific universal polymerase chain reaction (PCR) primers and a 16 S rRNA degenerate forward and reward primer provided in the procedure checklist document on the manufacturer’s website [24]. These primers were selected because of their ability to target a wide range of bacterial species. The amplification process was carried out following the manufacturer’s instructions for the PCR kit. In addition, the constructing amplicon libraries were generated using the SMRTbell^®^ Prep Kit 3.0 (Pacific Biosciences of California, Menlo Park, CA, USA). Amplicons were barcoded during either the PCR or library preparation using the SMRTbell barcoded adapters. Once amplification was complete, the final products were purified using AMPure PB beads (Pacific Biosciences of California). To quantify the amount of DNA obtained from the purification step, we applied the Qubit 2.0 fluorometer using the Qubit dsDNA HA Assay Kit (Thermo Fisher Scientific, Waltham, MA, USA). Finally, we performed sequencing on the PacBio Sequel ^®^ I system following manufacturer instructions.

### 2.4. Long-Read Sequencing (PacBio) Data Analysis

The quality of the sequencing data was checked using Fastqc and processed using the QIIME version qiime2 (version 2022.2; https://qiime2.org, accessed on 1 July 2024). Data were quality-filtered and dereplicated using DADA2 based on dada2 denoise-ccs. All sequences were performed in alignment with the representative sequences using Multiple Alignment via Fast Fourier Transform, and a phylogeny tree was constructed using FastTree. We classified each identical read and amplicon sequence variant to the highest resolution according to the SILVA database version 138 using “classify-sklearn”. Taxonomy bar plots were generated using the “barplot” alpha diversity metrics (observed features and the Shannon index). The beta diversity metric (weighted UniFrac) was calculated using “diversity core-metrics-phylogenetic” and “diversity beta-group-significance” based on their pairwise similarity. Linear discriminant analysis (LDA) and LDA Effect Size were performed according to operational taxonomic unit (OTU) characterizations at the genus level of the microbiota and grouping categories of AD, MCI, and healthy control.

## 3. Results

### 3.1. Participants and Clinical Status

A total of 100 participants (10 patients with AD, 46 patients with MCI, and 44 older adult controls) were included in the study. Table 1 shows the background and cognitive scores of each group. The average age of patients with AD, patients with MCI, and healthy controls was 66.90 years, 68.50 years, and 64.73 years, respectively. Age did not differ significantly among groups. In the AD group, the average Clinical Dementia Rating (CDR), Montreal Cognitive Assessment (MoCA), and Mini-mental Status Examination (MMSE) scores were 1.45, 13.9, and 16.7, respectively. Those of the MCI group were 0.41, 22.36, and 25.89, respectively.

### 3.2. Saliva Microbial Diversity in the AD, MCI, and Control Groups

After quality trimming and filtering, the valid unique tags that remained in the AD, MCI, and older adult control groups were 32,425, 27,309, and 26,583, respectively (Table 2). The number of OTUs was similar to the number of unique tags of each group. The unique tags and number of OTUs were significantly higher in the AD group than in the MCI and control groups (*p* < 0.0001). The alpha diversity, as measured by Shannon indices, showed that the oral microbiota diversity in the AD group was higher than that in the MCI and control groups. The alpha rarefaction curve was plotted as the number of OTUs versus the number of sequences to estimate the species richness of the saliva microbiota (Figure 1). The overall saliva microbiota diversity in the AD group tended to be higher than that in the MCI and control groups.

### 3.3. Taxonomic Classification of OTUs at the Phylum, Order, Family, and Genus Levels

Figure 2 illustrates the predominant taxa identified at the phylum, class, order, and family levels. At all levels (Figure 2A–D, respectively), we observed a significantly higher level of Fusobacteriota in the AD group than in the MCI and control groups. 

The distributions of the different microorganisms for the AD, MCI, and control groups are shown in Figure 3. At the phylum level, there was a higher level of Cyanobacteria in the control group than in the MCI and AD groups (Figure 3A). Furthermore, at the order level, Pseudomonadales were more enriched in the control group than in the MCI and AD groups (Figure 3C).

There was a higher level of Fusobacteriota at all taxonomic levels in the AD group than in the MCI and control groups. Moreover, at the family level, Peptostreptococcaceae was more enriched in the AD group than in the MCI and control groups (Figure 4C). In contrast, the level of Veillonella was lower in the AD group than in the control and MCI groups (Figure 4D).

## 4. Discussion

Our objective was to assess the comprehensive coverage of a full-length 16 S rDNA PacBio SMRT sequence for the bacterial composition of saliva in older adults diagnosed with AD and MCI. The use of the PacBio SMRT sequencing platform enabled us to generate read lengths exceeding 10 kb, which facilitated the identification of a broader range of bacterial taxa at various taxonomic levels. This approach offered more robust and accurate insight into the oral microbial community than conventional short-read sequencing methods. Moreover, it provided a rapid means of distinguishing oral bacterial profiles among AD patients, MCI patients, and healthy controls. However, it is worth noting that this method may be more costly than alternative diagnostic tests. In addition, the other generative molecular genetic techniques may be used to identify microbiome markers and might be helpful in defining the risk of AD and delaying the progression of Alzheimer’s disease.

We observed significantly higher levels of certain microbiota, including Peptostreptococcaceae and Fusobacteriaceae, in patients with AD, which is inconsistent with a previous study that reported lower Peptostreptococcaceae levels in AD patients than neurotypical controls [25]. The research study has shown that alterations in diet can lead to shifts in the diversity, composition, and stability of the gut microbiota [16]. Moreover, the previous study demonstrated that the probiotic treatment decreased relative abundance of Bacteroidetes (Prevotellaceae and Bacteroidaceae) while increasing Peptosteptococcaceae in patient samples [26]. Therefore, the discrepancy between our data and those of the previous study may result from the different diets in study populations. Nevertheless, this suggests a link between Peptostreptococcaceae and neurodegenerative diseases, such as AD. Studies have demonstrated the significance of this bacterial family in various conditions, such as type 2 diabetes [27], Crohn’s disease [28], metabolic diseases [29], and Clostridium difficile infections [30]. Specifically, the Peptostreptococcaceae family has been identified as part of the gut microbiota in individuals with type 2 diabetes and is associated with anti-inflammatory properties in the context of Crohn’s disease [28]. Peptostreptococcaceae, a family of bacteria present in the gut microbiome, plays a vital role in human health. These Gram-positive, obligate anaerobes are important members of normal human flora [31]. Research indicates that these bacteria are involved in the fermentation of dietary fibers, producing short-chain fatty acids (SCFAs) that are essential for colonic health and have anti-inflammatory properties [32]. The alteration of Peptostreptococcaceae found in AD samples may relate to the abnormal anti-inflammatory functions seen in the bodies of our population.

Fusobacteriaceae has also been linked to cognitive impairment. Studies have shown that Fusobacteriaceae play a role in the development of periodontal diseases, which have been associated with cognitive decline and neurodegenerative disorders [33]. Chronic stress, which increases the risk of developing MCI and AD, negatively impacts the hippocampus, a key brain region affected by AD, and may also influence gut microbiota composition, including Fusobacteriaceae [34]. Fusobacteriaceae, a family of anaerobic bacteria, play significant roles in both health and disease. The research study indicates that these bacteria are integral to the human gut microbiome, contributing to metabolic processes and immune modulation. For instance, they are involved in the fermentation of dietary fibers, producing short-chain fatty acids that are beneficial for gut health and which may influence systemic inflammation [35]. Furthermore, studies suggest that these bacteria can interact with host immune responses, potentially exacerbating inflammatory conditions [36]. Thus, the changes in Fusobacteriaceae observed in our AD sample may be linked to abnormal immune responses and inflammatory processes occurring within our population

We observed different levels of microbiota in the AD group, which included Proteobacteria, Pasteurellaceae, and Clostidia. Previous studies have reported an association between the level of Proteobacteria and the onset and progression of AD. Additionally, those with mild to moderate AD have been shown to have a significantly higher abundance of Proteobacteria, which suggests a link between this bacteria and cognitive decline [37]. 

Furthermore, dysbiosis characterized by an increase in Proteobacteria has been associated with inflammatory diseases, which could have implications for neurological health. Alterations in the gut microbiota, including a higher proportion of Proteobacteria, have been linked to various diseases, particularly those involving inflammation [38]. Some studies have also shown a correlation between an increase in Proteobacteria and cognitive impairment [39]. The gut–brain axis has been proposed as a mechanism through which Proteobacteria impacts cognitive function and neurological health. The gut microbiota metabolizes compounds that affect the gut–brain axis, which may be the underlying source of neurological diseases [40]. Understanding the influence of Proteobacteria on cognitive and neurological health is crucial for unraveling disease mechanisms and developing targeted interventions for these complex conditions. Additionally, studies have shown that the use of proton pump inhibitors increases the level of the Pasteurellaceae family in the gut microbiome, which is known to affect various aspects of health, including cognitive function [41]. Moreover, the study conducted by Karamujić-Čomić and colleagues found that the class Clostridia, order Clostridiales, family Christensenellaceae, and genus Christensenellaceae R7 group showed a higher abundance in individuals with better cognition [42]. This finding suggests a potential link between the abundance of Clostridia and cognitive performance, and thus, cognitive health. 

We found that at the phylum level, Cyanobacteria were less abundant in the saliva sample of the AD group than in the MCI and control group samples. This is inconsistent with a previous study that reported a link between Cyanobacteria and neurodegenerative diseases, such as AD and Parkinson’s disease [43]. The presence of Cyanobacteria in the gut microbiota has been associated with the onset of neurodegenerative diseases, which indicates a role in the pathogenesis of such conditions [43]. Additionally, an imbalance in the gut microbiota due to an increase in Cyanobacteria has been associated with cognitive impairment and neurological disorders, and probiotics that modulate the gut microbiota, including Cyanobacteria, can enhance cognitive function and alleviate the symptoms of neurological disorders [44]. These findings demonstrate that the composition of the gut microbiota, including Cyanobacteria, has a significant impact on brain development and cognitive function.

In contrast to microbes, such as Fusobacterium, the level of the Veillonella genus was lower in the AD group than in the MCI and control groups. This genus consists of anaerobic, Gram-negative cocci, and has been implicated in various aspects of human health and disease. Veillonella contributes to oral biofilm ecology and may act as an accessory pathogen that promotes the growth of pathogenic species within biofilms. The ability of Veillonella to prevent the growth of other bacteria, such as *Fusobacterium nucleatum*, in microaerophilic environments further highlights its importance in oral health [45].

Recent research has demonstrated that there are alterations in the oral microbiome composition in individuals with AD [46]. The oral microbiome has been proposed to play a role in the pathophysiology of various mental disorders, including AD [47]. The most diverse communities of microbiota are found in the gut, followed by the mouth [48]. In the present study, a higher diversity of the oral microbiota indicated greater dysbiosis in the oral cavity in patients with AD than in patients with MCI and controls. Dysbiosis of the oral microbiota has been increasingly recognized as a significant factor in the pathogenesis and progression of various diseases, including non-alcoholic fatty liver disease [49], valvular heart disease [50], systemic inflammation, immunoreaction, oxidative stress, and thrombosis [51].

The previous studies demonstrated that high-throughput long read next generation sequencing platform, such as the PacBio system, despite delivering a smaller volume of clean data compared to the MiSeq platform, exceeded the resolution of this platform by offering longer read lengths and more detailed annotations of nucleotide sequences down to the species or strain level [52]. Moreover, Buetas and their team reported that oral samples sequenced with Illumina and PacBio technologies are mostly comparable. However, PacBio reads provided more accurate species-level assignments compared to Illumina [53]. This suggests that PacBio technology is advantageous for use in future microbiome studies, even though it currently involves higher costs to achieve a similar number of reads per sample.

Recent research has showed a causal relationship between microbiomes and the development of central nervous system (CNS) disorders in Alzheimer’s disease (AD). The microbiome significantly influences both immune responses and the balance within the oral cavity [54]. The oral cavity is covered by a microbial biofilm on all biological and non-biological surfaces, making the regulation of interactions between the host and these microorganisms crucial for maintaining healthy teeth and implants [55]. Beyond the gut, the oral cavity is home to a wide variety of bacteria that interact with each other and with the host. These oral bacteria can enter the bloodstream through activities like tooth brushing or gum inflammation, potentially impacting peripheral organs and the CNS. Although the oral cavity is home to many commensal bacteria, it is still uncertain whether particular bacteria directly contribute to triggering harmful brain responses. However, evidence suggests that bacteria present in the human oral cavity can encourage Aβ formation and play a role in the development of Alzheimer’s disease [56]. Moreover, recent studies suggest a possible connection between *Fusobacterium nucleatum* (*F. nucleatum*) and the development and progression of Alzheimer’s disease. Some studies have identified that *F. nucleatum* produces an adhesion protein called FadA under stressful and unhealthy conditions [57]. This amyloid-like FadA may act as a scaffold for biofilm formation and is resistant to acidic environments. These findings illuminate how *F. nucleatum* contributes to the creation of a stable amyloid-like structure. Additionally, *F. nucleatum* infection has been associated with increased systemic or local inflammation, which can compromise the blood–brain barrier. This disruption leads to systemic and central nervous system (CNS) immune activation, neuroinflammation, the accumulation of amyloid-beta plaques in the brain, and the progression of AD. Therefore, our study showed that the significant increase in Fusobacterium in all levels of sample from AD patients compared to the control group may relate to the previous evidence, with similar mechanisms or hypotheses action. In addition, some studies have proposed five pathways that relate to the accumulation of key pathogen factors, amyloid beta and Tau, and suggest how oral pathogenic microbiomes might contribute to the onset and progression of Alzheimer’s disease (AD) [56]. 

There are some limitations that could be improved in our study. Currently, the PacBio platform is more expensive than other tests, but increased demand could lead to reduced costs and greater accessibility for this technology. Moreover, our samples mostly came from patients in the central region of Thailand, which also limits the number of AD patients. Therefore, future studies using the high-throughput PacBio platform should analyze the samples from different regions, and a greater number of AD samples might provide more complete information on the microbiome that relates to this disease. Interestingly, our results show similar trends in the microbiome alterations in samples from patients with AD and MCI in both groups, suggesting that some microbiome biomarkers can predict the risk of a disease status change from MCI to AD. Targeted approaches to altering the oral microbiota might provide a promising strategy for delaying the onset of Alzheimer’s disease, or slowing its progression.

## 5. Conclusions

Our study represents a pioneering effort in using the PacBio SMRT sequencing platform to investigate the relationship between oral microbiota and the neurodegenerative diseases of AD and MCI. Our results demonstrate that this advanced sequencing platform has the potential to identify possible biomarkers for AD. Notably, patients with AD exhibited higher microbial diversity than those with MCI and healthy controls. Furthermore, significantly higher levels of Peptostreptococcaceae and Fusobacteriaceae were observed in individuals with AD, which suggests an association between these bacterial taxa and the AD pathology. Our study underscores the potential use of PacBio SMRT sequencing in unraveling the intricate interplay between oral microbiota composition and neurodegenerative conditions, such as AD. Moreover, our findings offer valuable insight into the future of diagnostic and therapeutic strategies. 

## Figures and Tables

**Figure 1 ijerph-21-01242-f001:**
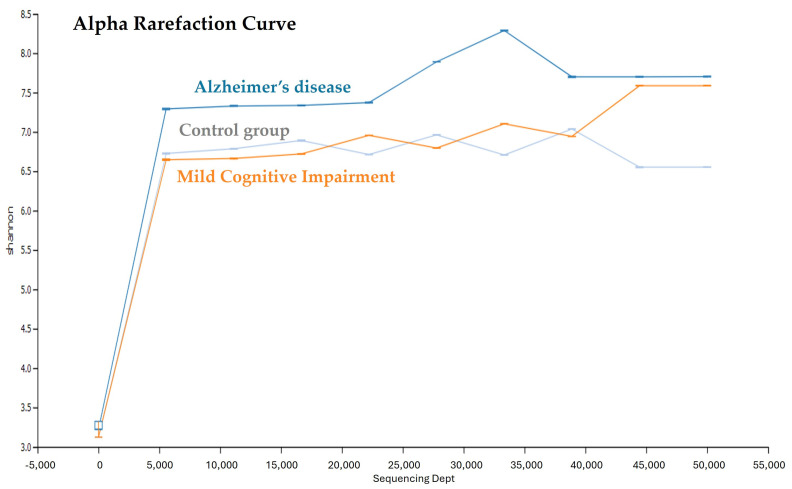
Alpha rarefaction curves used to estimate the Shannon indices of the oral microbiota of the AD, MCI, and control groups.

**Figure 2 ijerph-21-01242-f002:**
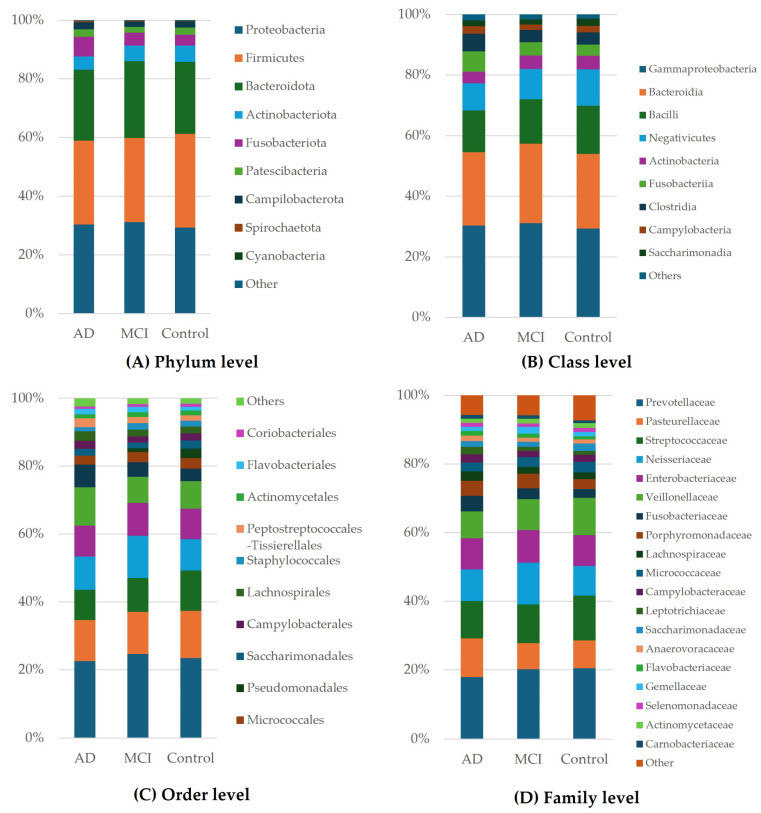
Bar plots of the average bacterial profile of the OTUs of saliva in the oral microbiota of the AD, MCI, and control groups at the (**A**) phylum level, (**B**) class level, (**C**) order level, and (**D**) family level.

**Figure 3 ijerph-21-01242-f003:**
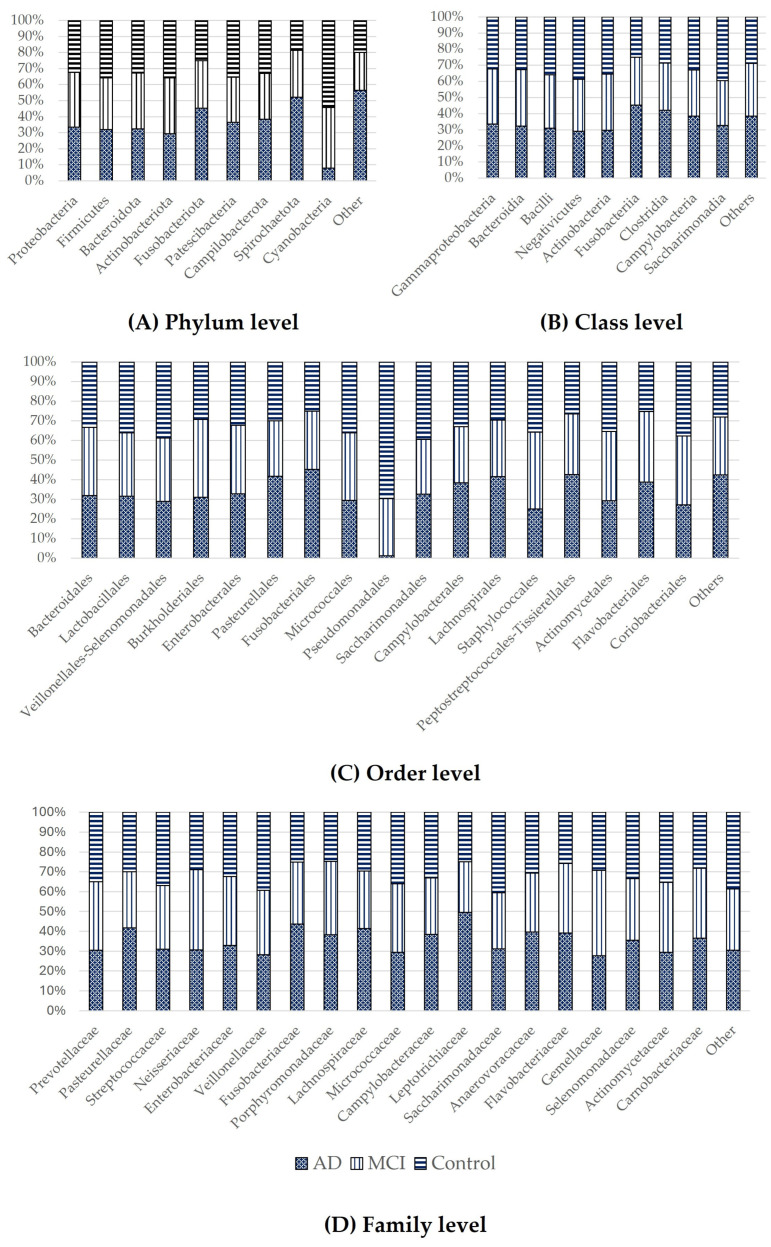
Bar plots of the average proportion of the different microbiota in the saliva of the AD, MCI, and control groups at the (**A**) phylum level, (**B**) class level, (**C**) order level, (**D**) and family level.

**Figure 4 ijerph-21-01242-f004:**
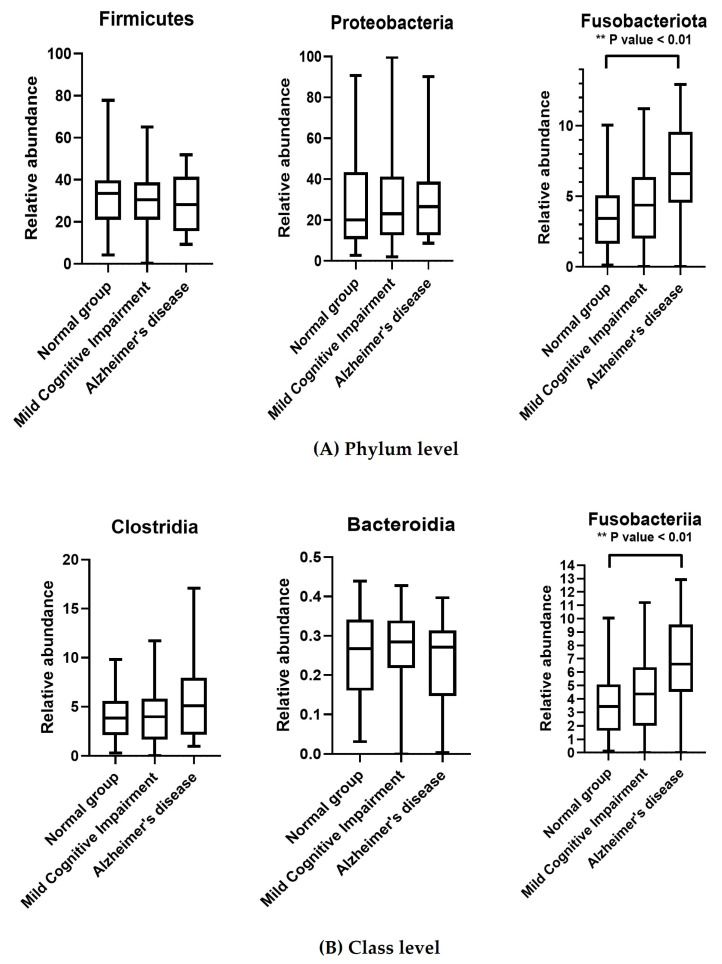
Comparisons of the relative abundance of oral bacteria at the (**A**) phylum, (**B**) class, (**C**) genus, and (**D**) family levels among the AD, MCI, and control groups. * *p* < 0.05 and ** *p* < 0.01, One-Way ANOVA and multiple comparison. Relative abundance demonstrates what percentage of the microbiome is made up of a specific organism.

**Table 1 ijerph-21-01242-t001:** Characteristics of patients with Alzheimer’s disease (AD), mild cognitive impairment (MCI), and older adult controls.

Basic Profile	AD	MCI	Control
Sample size, N	10	46	44
Male:female	4:6	14:32	12:32
Age, years	66.90 ± 7.06	68.50 ± 6.35	64.73 ± 4.78
CDR score	1.45 (0.5–2)	0.41 (0–1)	n/a
MoCA score	13.90 (5–19)	22.36 (16–27)	26.91 (20–30)
MMSE score	16.70 (12–21)	25.89 (20–29)	28.14 (24–30)

CDR: Clinical Dementia Rating, MoCA: Montreal Cognitive Assessment, MMSE: Mini-mental Status Examination.

**Table 2 ijerph-21-01242-t002:** Number of unique tags, operational taxonomic units (OTUs), and alpha diversity estimates of oral bacterial communities in AD patients, MCI patients, and older adult controls.

Group	AD	MCI	Control
Unique tags	32,425 ± 17,819 *	27,309 ± 11,052	26,583 ± 6887
No. of OTUs	31,105 ± 17,441 *	25,641 ± 10,610	25,156.7 ± 7182
Richness index	60.8 ± 22.56	50.67 ± 14.77	50.0 ± 11.75
Diversity index (Shannon)	7.04 ± 1.42	6.48 ± 1.22	6.54 ± 1.09

* *p* < 0.0001. One sample *t*-test between AD and MCI or control groups.

## Data Availability

The datasets used and analyzed during the current study are available from the corresponding author on reasonable request.

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
