# Peer review of "Analysis of Oral Microbiota in Elderly Thai Patients with Alzheimer’s Disease and Mild Cognitive Impairment"

_ijerph, 2024, doi:10.3390/ijerph21091242_

Round 1
Reviewer 1 Report
Comments and Suggestions for Authors
Title: Analysis of oral microbiota in elderly Thai patients with Alzheimer’s disease and mild cognitive impairment.
Summary
In this research article, Narongrit Sritana et al. investigated the relationship between oral microbiota composition and a neurodegenerative disease, Alzheimer’s disease (AD). The relative abundance and diversity of bacterial taxa present in saliva among older adults diagnosed with AD, those with mild cognitive impairment (MCI), and healthy controls were evaluated and analyzed in this study. Saliva samples and clinical data were collected from 10 AD patients, 46 MCI patients, and 44 healthy older adults. The data indicated that AD patients exhibited significantly higher levels of Fusobacteriota and Peptostreptococcaceae and lower levels of Veillonella than the MCI and control groups. It is concluded that a high abundance of Fusobacteria at various levels may serve as a biomarker for AD. However, this manuscript is not publishable in the International Journal of Environmental Research and Public Health without major revision.
Major and minor comments are listed below.
1. all the names of bacterial species in this manuscript should be in italics;
2. -page 10, lines 263-265, “We observed significantly higher levels of certain microbiota, including Peptostreptococcaceae and Fusobacteriaceae, in patients with AD, which is inconsistent with a previous study that reported lower Peptostreptococcaceae levels in AD patients than neurotypical controls [21]. ”
What are the explanations for this discrepancy?
3. 10 AD patients, 46 MCI patients, and 44 healthy older adults were included in this study.
Are these patients recruited in the same hospital or institute?
More AD patients should be recruited. The number of AD patients should be comparable to that of MCI patients.
4. To highlight the differences between Figure 2 and Figure 3, try to use a different visualization method to present Figure 3.
5. Figure 4, what is the definition of relative abundance?
6. Why did the Peptostreptococcaceae and Fusobacteriaceae species show different levels between AD patients and control groups? What are their physiological functions?
7. -lines 334-336, “Our results demonstrated that this advanced sequencing platform has the potential to identify precise biomarkers for AD. Notably, patients with AD exhibited higher microbial diversity than those with MCI and healthy controls.”
The authors should apply this sequencing platform to detect the microbial levels of other samples from a different region or a different group of patients to demonstrate this technique is robust or these biomarkers are valid.
Author Response
- all the names of bacterial species in this manuscript should be in italics;
Response: Thank you for pointing this out. We agree with this comment. Therefore, we found the bacterial species only in the introduction part (Lactobacillus rhamnosus) [The change on page number 2 at line No. 76] also one in the reference (Faecalibacterium prausnitzii) [The change on page number 13 at line No. 491] that we’d already changed to italics style as your comment.
- -page 10, lines 263-265, “We observed significantly higher levels of certain microbiota, including Peptostreptococcaceae and Fusobacteriaceae, in patients with AD, which is inconsistent with a previous study that reported lower Peptostreptococcaceae levels in AD patients than neurotypical controls [21]. ”
What are the explanations for this discrepancy?
Response: We add more information in the discussion part of the article as the following context
" The research study has shown that alterations in diet can lead to shifts in the diversity, composition, and stability of the gut microbiota [16]. Moreover, the previous study demonstrated that the probiotic treatment decreased relative abundance of Bacteroidetes (Prevotellaceae and Bacteroidaceae), while increasing Peptosteptococcaceae in patient samples [26]. Therefore, the discrepancy between our data and the previous study may occur from the different diets in study populations."
This change can be found on page number 10 at line No. 276-281.
- 10 AD patients, 46 MCI patients, and 44 healthy older adults were included in this study.
Are these patients recruited in the same hospital or institute?
More AD patients should be recruited. The number of AD patients should be comparable to that of MCI patients.
Response: The saliva samples were collected from 3 hospital regions, Chulabhorn Hospital, King Chulalongkorn Memorial Hospital and Ramathibodi Hospital. We also add more details and suggestion idea in material and methods and discussion part of the article as the following details;
"Saliva samples were collected from 100 older adult native Thai participants who had visited Chulabhorn Hospital, Ramathibodi Hospital and King Chulalongkorn Memorial Hospital."
This change can be found on page number 3 at line No. 132-133.
********
"There are some limitations that could be improved for our study. Currently, the PacBio platform is more expensive than other tests, but increased demand could lead to reduced costs and greater accessibility for this technology. Moreover, our samples mostly came from patients in central region of Thailand also limit number of AD patients. Therefore, the future study using high throughput Pacbio platform analyze the samples from different regions also a greater number of AD samples might provide the complete information of microbiome that relate to this disease."
This change can be found on page number 11 at line No. 371-377.
- To highlight the differences between Figure 2 and Figure 3, try to use a different visualization method to present Figure 3.
Response: We change the figure 3 to new visualization image and add into the article.
This change can be found on page number 7 at line No. 254.
- Figure 4, what is the definition of relative abundance?
Response: Relative abundance demonstrates how many percentages of the microbiome are made up of a specific organism, for example in figure 4C the Fusobacterium present about 4% of the total amount of bacteria detected in the sample from AD patients.
This change can be found on page number 9 at line No. 261-262.
- Why did the Peptostreptococcaceae and Fusobacteriaceae species show different levels between AD patients and control groups? What are their physiological functions?
Response: According to limit information of our literature review we found that Peptostreptococcaceae may relate to anti-inflammatory function in human body and Fusobacteriaceae may be linked to abnormal immune responses and inflammatory processes. We also add this additional context in to the article as followed.
"Peptostreptococcaceae, a family of bacteria present in the gut microbiome, plays a vital role in human health. These gram-positive, obligate anaerobes are important members of the normal human flora [31]. Research indicates that these bacteria are involved in the fermentation of dietary fibers, producing short-chain fatty acids (SCFAs) that are essential for colonic health and have anti-inflammatory properties [32]. The alteration of Peptostreptococcaceae that found in AD sample may relate to abnormal anti-inflammatory function occur in the body of our population."
This change can be found on page number 10 at line No. 287-294.
"Fusobacteriaceae, a family of anaerobic bacteria, play significant roles in both health and disease. The research study indicates that these bacteria are integral to the human gut microbiome, contributing to metabolic processes and immune modulation. For instance, they are involved in the fermentation of dietary fibers, producing short-chain fatty acids that are beneficial for gut health and may influence systemic inflammation [35]. Furthermore, studies suggest that these bacteria can interact with host immune responses, potentially exacerbating inflammatory conditions [36]. Thus, the changes in Fusobacteriaceae observed in our AD sample may be linked to abnormal immune responses and inflammatory processes occurring within our population."
This change can be found on page number 10 at line No. 300-309.
- -lines 334-336, “Our results demonstrated that this advanced sequencing platform has the potential to identify precise biomarkers for AD. Notably, patients with AD exhibited higher microbial diversity than those with MCI and healthy controls.”
The authors should apply this sequencing platform to detect the microbial levels of other samples from a different region or a different group of patients to demonstrate this technique is robust or these biomarkers are valid.
Response: We change some of context according to your comment and provide more information and references publication that use similar platform to our article as the following details;
"The previously studies demonstrated that high throughput long read next generation sequencing platform, the PacBio system, although delivering a smaller volume of clean data compared to the MiSeq platform, exceeded its resolution by offering longer read lengths and more detailed annotation of nucleotide sequences down to the species or strain level [52]. Moreover, Buetas and teams reported that oral samples sequenced with Illumina and PacBio technologies are mostly comparable. However, PacBio reads provided more accurate species-level assignments compared to Illumina [53]. This suggests that PacBio technology is advantageous for future microbiome studies, even though it currently involves higher costs to achieve a similar number of reads per sample."
This change can be found on page number 11 at line No. 362-370.
Reviewer 2 Report
Comments and Suggestions for Authors
Dear Authors,
My only remarks are as following:
1.Authors should specify what bacteria specific primer has been utilized by them.
2. Figure 1 quality should be improved, especially text associated with it.
3. In my opinion it would be fitting to inform the readers that in the introduction section, microbiome composition changes that are tied to dietary changes, are ,at best, short-lived.
4. The phrase "normal group(s)" should be changed to "control group(s)" in the whole body of the manuscript, as this difference in naming, might be confusing for the readers.
Both from the side of methodology, and execution the uploaded manuscript meets, the quality standards that allow for its publishing. References are complete and properly formatted. The paper is introducing some novelty in its area. In the results presented by the authors, there are some inconsistencies with the currently present literature (and most of the literature chosen by authors has been properly reviewed, so I don’t see any problem with it), which will need further research on the topic to be confirmed. As stated in the initial remarks, besides minor editorial changes, the article at this point, in my opinion, does meet the standards to be published in a scientific journal. In case of the conclusions, some of the findings need further confirmation, but the presented results might be regarded as at least "interesting" for the field.
I hope that those small remarks will help you in improving your proposed manuscript.
Comments on the Quality of English LanguageIn my opinion, the proposed manuscript has met the standards needed for publication in a scientific journal.
Author Response
1.Authors should specify what bacteria specific primer has been utilized by them.
Response: We added the details about primer set in the materials and method of the article as following details;
"that provide as procedure checklist document from manufacturer website [24]."
This change can be found on page number 4 at line No. 166-167.
- Figure 1 quality should be improved, especially text associated with it.
Response: We'd already changed to new version of figure 1 and add into the article as your comment and upload the original image to the journal.
This change can be found on page number 6 at line No. 247.
- In my opinion it would be fitting to inform the readers that in the introduction section, microbiome composition changes that are tied to dietary changes, are ,at best, short-lived.
Response: Thank you for your suggestion. We add the information according to your comment in the introduction part of the article as following details;
" In addition, the composition of both the gut and oral microbiomes can be significant-ly influenced by dietary changes. Research has shown that alterations in diet can lead to shifts in the diversity, composition, and stability of the gut microbiota [16]. For instance, major dietary shifts, such as increased intake of processed carbohydrates, have been asso-ciated with significant changes in the diversity and composition of the oral microbiome [17]. Moreover, dietary interventions have been found to impact the gut microbiome, with changes in the gut microbiome composition explaining a portion of the variance in serum metabolites in response to diet [18]."
This change can be found on page number 2 at line No. 88-95.
- The phrase "normal group(s)" should be changed to "control group(s)" in the whole body of the manuscript, as this difference in naming, might be confusing for the readers.
Response: Thank you for your suggestion. We add the information according to your comment in all part of the article.
This change can be found on page number 6 at line No. 247.
Both from the side of methodology, and execution the uploaded manuscript meets, the quality standards that allow for its publishing. References are complete and properly formatted. The paper is introducing some novelty in its area. In the results presented by the authors, there are some inconsistencies with the currently present literature (and most of the literature chosen by authors has been properly reviewed, so I don’t see any problem with it), which will need further research on the topic to be confirmed. As stated in the initial remarks, besides minor editorial changes, the article at this point, in my opinion, does meet the standards to be published in a scientific journal. In case of the conclusions, some of the findings need further confirmation, but the presented results might be regarded as at least "interesting" for the field.
I hope that those small remarks will help you in improving your proposed manuscript.
Comments on the Quality of English Language
In my opinion, the proposed manuscript has met the standards needed for publication in a scientific journal.
